# SARS-CoV-2 Nucleocapsid Protein Is Not Responsible for Over-Activation of Complement Lectin Pathway

**DOI:** 10.3390/ijms25137343

**Published:** 2024-07-04

**Authors:** Andrea Kocsis, Dalma Bartus, Edit Hirsch, Mihály Józsi, István Hajdú, József Dobó, Júlia Balczer, Gábor Pál, Péter Gál

**Affiliations:** 1Institute of Molecular Life Sciences, HUN-REN Research Centre for Natural Sciences, Hungarian Research Network, H-1117 Budapest, Hungary; kocsis.andrea@ttk.hu (A.K.);; 2Department of Organic Chemistry and Technology, Faculty of Chemical Technology and Biotechnology, Budapest University of Technology and Economics, H-1111 Budapest, Hungary; hirsch.edit@vbk.bme.hu; 3Department of Immunology, ELTE Eötvös Loránd University, H-1117 Budapest, Hungary; 4HUN-REN-ELTE Complement Research Group, Hungarian Research Network, H-1117 Budapest, Hungary; 5Department of Biochemistry, ELTE Eötvös Loránd University, H-1117 Budapest, Hungary

**Keywords:** complement lectin pathway, MASP-2, COVID-19, SARS-CoV-2 nucleocapsid protein, proteolytic activity, zymogen activation

## Abstract

The nucleocapsid (N) protein of severe acute respiratory syndrome coronavirus 2 (SARS-CoV-2) is a viral structural protein that is abundant in the circulation of infected individuals. Previous published studies reported controversial data about the role of the N protein in the activation of the complement system. It was suggested that the N protein directly interacts with mannose-binding lectin-associated serine protease-2 (MASP-2) and stimulates lectin pathway overactivation/activity. In order to check these data and to reveal the mechanism of activation, we examined the effect of the N protein on lectin pathway activation. We found that the N protein does not bind to MASP-2 and MASP-1 and it does not stimulate lectin pathway activity in normal human serum. Furthermore, the N protein does not facilitate the activation of zymogen MASP-2, which is MASP-1 dependent. Moreover, the N protein does not boost the enzymatic activity of MASP-2 either on synthetic or on protein substrates. In some of our experiments, we observed that MASP-2 digests the N protein. However, it is questionable, whether this activity is biologically relevant. Although surface-bound N protein did not activate the lectin pathway, it did trigger the alternative pathway in 10% human serum. Additionally, we detected some classical pathway activation by the N protein. Nevertheless, we demonstrated that this activation was induced by the bound nucleic acid, rather than by the N protein itself.

## 1. Introduction

The coronavirus disease 19 (COVID-19) pandemic caused by the severe acute respiratory syndrome coronavirus 2 (SARS-CoV-2) put a significant burden on societies and economies around the globe. The SARS-CoV-2 infection may result in severe clinical symptoms, such as acute respiratory failure, circulatory shock, acute renal failure, and thrombotic complications. Innate immunity is responsible for the early response against viral infections. The complement system is an ancient effector arm of the innate immune system [1]. It is a proteolytic cascade system that can be triggered by various stimuli. The complement system can be activated through three different pathways: the classical pathway (CP), the lectin pathway (LP), and the alternative pathway (AP). The CP is antibody dependent, while the LP and the AP can be triggered by various activating surfaces. The lectin pathway is one of the first lines of defense against invading pathogens. Pattern recognition molecules (PRMs) (mannose-binding lectin (MBL), ficolins, collectins) in the circulation bind to the pathogen-associated molecular patterns (PAMPs) and MBL-associated serine proteases, and the so-called mannose-binding lectin-associated serine protease-1 and -2 (MASP-1 and MASP-2) initiate a proteolytic cascade reaction [2]. Activation of the complement system leads to the opsonization and lysis of the attacked (e.g., virus infected) cells, and the anaphylatoxins released by the proteolytic cascade initiate inflammation. MASP-1 and MASP-2 are equally important enzymes for LP activation [3]. The first enzymatic step during LP activation is the autoactivation of zymogen MASP-1. Activated MASP-1 then cleaves and activates zymogen MASP-2. Active MASP-2 is the only enzyme in the LP that—by cleaving C4—generates the C4b component of the C4b2b C3 convertase complex. The other component (C2) is cleaved by both MASP-1 and MASP-2.

An increasing body of evidence indicates that the LP activates during SARS-CoV-2 infection [4,5]. Uncontrolled overactivation of the LP may contribute to the severe life-threatening symptoms of the coronavirus infection. Complement-mediated microvascular injury was detected in the lungs of patients with severe COVID-19. Immunohistochemistry analysis detected striking levels of MASP-2, C4d, and C5b-9 deposition in the infected tissues [6,7,8]. Treatment of COVID-19 patients with Narsoplimab, an inhibitory anti-MASP-2 antibody, had beneficial effect and strongly suggested that endothelial injury-induced activation of MASP-2 and the LP play a central role in the pathophysiology of COVID-19-related lung injury [9,10].

Based on the above evidence, there is a consensus in the literature that the LP is activated during SARS-CoV-2 infection; however, the mechanism is debated. SARS-CoV-2 is a single-stranded RNA virus encoding four structural proteins: spike (S), envelope (E), membrane (M), and nucleocapsid (N). It was shown that MBL binds to trimeric S protein and this binding initiates LP activation [10,11]. However, the data concerning the role of the N protein in LP activation are controversial. N protein is the most abundant viral protein in the circulation during coronavirus infection. Its role inside the virus is to cover and stabilize the RNA genome, forming the ribonucleoprotein core. It can also escape from the virus-infected cells, and it may stick to extracellular surfaces. There are contradictory data whether the N protein binds any PRMs of the lectin pathway [10,11,12]. Moreover, there are publications claiming that the N protein binds directly to MASP-2 and stimulates its enzymatic activity, causing LP overactivation [13,14]. These publications suggest that in this way N protein significantly contributes to the development of the severe inflammatory symptoms.

In order to clarify the role of the N protein in lectin pathway activation, we decided to examine its capacity to activate the LP and its interactions with LP proteases MASP-1 and MASP-2 to reveal the activation mechanism. We found that N protein is only a weak complement activator, which activates the AP, and to a lesser extent CP, but not the LP. We could not detect any interaction between the N protein and MASP-1 or MASP-2. The N protein did not stimulate the enzymatic activity of MASP-2. We concluded that the N protein itself is not responsible for the strong LP activation elicited by SARS-CoV-2 infection.

## 2. Results

### 2.1. Nucleic Acid-Free N Protein Does Not Elicit C4b Deposition in Normal Human Serum (NHS)

In order to check whether the surface-deposited N protein can activate the LP, we covered the wells of the microtiter plates with recombinant N protein and after incubation with 10% NHS, we detected C4b and C3b deposition. Since the biological function of the N protein is to bind the viral RNA, the N protein might carry nucleic acid in the circulation of the infected patients. We expressed the recombinant N protein in *E. coli* and purified two N protein forms: one that is nucleic acid-free and another one that is in complex with nucleic acids. The nucleic acid-free N protein did not trigger any C4b deposition in 10% NHS, indicating that there is no lectin (or classical) pathway activation (Figure 1). The nucleic acid-bound N protein elicited a weak C4b deposition, which could not be prevented by the specific MASP-2 inhibitor SGMI-2. This weak C4b deposition could therefore be due to CP activation. In contrast, both N protein forms triggered significant C3b deposition. The observed C3b deposition in the absence of C4b deposition clearly demonstrated that the N protein itself activates only the AP but not the LP or CP. The N protein is not glycosylated since it does not go through the secretory pathway. However, in some experiments, the authors used recombinant N protein expressed in mammalian cells, which might be glycosylated [10]. To rule out the possibility that the discrepancy between the experimental results is due to glycosylation of the recombinant N protein of mammalian origin, we tested the complement-activating ability of recombinant N protein expressed in HEK cells. We used the same experimental setting as in the case of the recombinant N protein of bacterial origin. However, we did not obtain significant C4b deposition in this case either (Appendix A). We can conclude that regardless of its origin, surface-bound recombinant N protein cannot activate the LP.

### 2.2. The N Protein Does Not Stimulate the Enzymatic Activity of MASP-2

Previous reports suggested that the N protein can boost the catalytic activity of MASP-2 [13,14]. In our experiments, we checked the effect of the N protein on the enzymatic activity of MASP-2 towards synthetic and protein substrates. In our experiments, we used the recombinant catalytic fragment of MASP-2 containing the two complement control proteins (CCPs) and the serine protease (SP) domains (CCP1-CCP2-SP). The same construct was used by the previous studies in the literature [13,14]. Our recombinant MASP-2 exhibited high catalytic activity on both the synthetic (Z-Gly-Gly-Arg-AMC) and the natural (C4) substrates. In the case of the synthetic substrate digestion, we used the same experimental conditions as Kang et al. [14]. Under these conditions, our MASP-2 protein cleaved the synthetic substrate completely in a few minutes. Addition of the N protein in high concentration (8–16 µM) could not further enhance the catalytic activity of MASP-2 (Figure 2A). The N protein preparation did not cleave the synthetic substrate at all, which means that it was free from any proteolytic contamination. We also tested the catalytic activity of a stable zymogen mutant (Arg444Gln) form of MASP-2 [15]. During zymogen activation, trypsin-like serine proteases undergo limited proteolysis resulting in the cleavage of an Arg-Ile bond in the activation peptide followed by a conformational change. In our MASP-2 mutant, the corresponding Arg444 is mutated to a cleavage-resistant glutamine, which stabilizes the protein in its one-chain zymogen form. Since the catalytic triad (Asp, His, Ser) is intact in the one-chain form, this molecule may adopt an active-like conformation exhibiting enzymatic activity. In spite of that, the zymogen mutant MASP-2 did not cleave the synthetic substrate even in the presence of N protein at high (8–16 µM) concentrations (Figure 2B). This demonstrates that the N protein cannot facilitate any zymogen-to-active conformational change in zymogen MASP-2.

In the next experiment, we tested whether the proteolytic activity of MASP-2 is affected by the N protein. We used both the zymogen and the active form of MASP-2 and followed the cleavage of the physiological substrate, C4. The active MASP-2 cleaved C4 with high efficiency even at a low concentration (2nM) (Figure 3A,B), but the zymogen also showed some proteolytic activity at higher concentration (3µM) (Figure 3C,D). The large C4 molecule makes many contacts with the MASP-2 molecule and fosters the zymogen-to-active conformational change in the one-chain form [15,16]. The N protein was not able to affect the proteolytic activity of either the active or the zymogen form.

In summary, our results prove that the N protein cannot stimulate the catalytic activity of MASP-2. On the other hand, during this experiment, we detected that MASP-2 cleaves the N protein. The intensity of the band corresponding to the N protein was diminished gradually on the SDS PAGE with the progression of the proteolytic reaction. This result indicates that the N protein is a potential substrate of the MASP-2 protease. The MASP-2-mediated proteolysis of the N protein was corroborated in a separate experiment, where MASP-2 and the N protein were incubated together at 37 °C and followed by SDS PAGE (Appendix A).

### 2.3. The N Protein Does Not Stimulate the Proteolytic Activity of the MBL–MASP-2 Complexes

In the previous experiments, we mixed the isolated proteins in the fluid phase and showed that the N protein cannot stimulate the enzymatic activity of MASP-2. However, LP activation usually takes place on an activation surface and MASP-2 is in a complex with PRMs, such as MBL. This scenario is much closer to the physiological situation than the fluid phase reaction between the MASP-2 catalytic fragment and the N protein. To test whether the N protein has any effect on the proteolytic activity of the surface-deposited MBL–MASP-2 complexes, we used the assay system developed by Petersen et al. [17]. In this assay system, the MBL–MASP-2 complexes are deposited on a mannan-coated surface and the components of the CP are washed away using a high salt concentration (1 M NaCl). After washing, we added C4 to the surface-deposited MBL–MASP-2 complexes with or without N protein. After incubation at 37 °C for 0–70 min, we detected the C4b deposition at different time points (Figure 4). The presence of the N protein did not make any difference in the C4b deposition. We can conclude that even when MASP-2 is in complex with MBL, the N protein cannot stimulate its proteolytic activity.

### 2.4. The N Protein Does Not Stimulate LP Activation in the Presence of MASP-1-Specific Inhibitor

MASP-1 and MASP-2 are equally important enzymes for lectin pathway activation. Although in vitro experiments proved that isolated zymogen MASP-2 can autoactivate slowly, this phenomenon does not manifest in the blood. In human blood, MASP-1 is the exclusive activator of MASP-2. Since MASP-1 activates MASP-2 very quickly and efficiently, even if the N protein stimulated the autoactivation of MASP-2, we most probably could not detect that due to the action of MASP-1. To check this possibility, we incubated 2% NHS on a mannan-coated surface in the presence of 2 µM N protein and our MASP-1-specific inhibitor, SGMI-1, which blocks the contribution of MASP-1 (Figure 5). Under these circumstances, we can study the effect of N protein on the MBL–MASP-2 complexes in the presence of all serum proteins but without the contribution of MASP-1. If the N protein were able to boost MBL-bound zymogen MASP-2 autoactivation, we could sensitively detect this effect in this system. However, we could not detect any N protein-mediated MASP-2 activation in the presence of SGMI-1. The inhibition of MASP-1 completely blocked the C4b deposition, and the presence or absence of the N protein did not make any difference. MASP-1 inhibition was as effective in blocking LP activation as MASP-2 inhibition in the presence of N protein.

### 2.5. The N Protein Does Not Bind to MASP-2 and MASP-1

A previous study using SPR reported direct interaction between the catalytic fragment of MASP-2 (CCP1-CCP2-SP) and the N protein [13]. We used a similar technique, BLI (bio-layer interferometry). We immobilized the recombinant N protein via N-terminal His-tag to the BLI sensor and incubated it in different protein solutions. To validate the system, we used long pentraxin 3 (PTX3), which was previously shown to bind to N protein with high affinity [11]. In our experimental system, the PTX3 bound to the immobilized N protein with a K_D_ of (1.3 ± 0.8) × 10^−8^ M (Figure 6). When we used the wild-type MASP-2 catalytic fragment, we obtained a negative signal. The reason for this unexpected phenomenon could be that the MASP-2 protease digests the immobilized N protein on the sensor (Appendix A). The same phenomenon was detected in the case of the Arg444Gln zymogen mutant, which also has some proteolytic activity on the N protein. In order to eliminate the effect of the proteolysis, we used the Ser633Ala mutant MASP-2. In this mutant the catalytic Ser633 in the active site is mutated to alanine. The Arg^444^Ile^445^ bond in the activation peptide was cleaved with MASP-1 [16]. In this way, we obtained a proteolytically inactive two-chain form of MASP-2. In the BLI experiment, however, this inactive form of MASP-2 was not able to bind to the N protein. We can conclude that the CCP1-CCP2-SP catalytic fragment of MASP-2 does not bind to the N protein. The negative results obtained with BLI are consistent with the above enzymatic and functional measurements and confirm that there is no interaction between the two proteins. Finally, we tested the binding of MASP-1 to the N protein. To avoid the proteolysis of the N protein, in this experiment we used the inactive catalytic fragment (CCP1-CCP2-SP, Ser646Ala) of MASP-1. Similar to MASP-2, we did not detect any binding to the N protein.

## 3. Discussion

It has been known since the appearance of the SARS-CoV-1 epidemic in 2002 that complement activation could contribute to viral pathogenesis. In an animal model, C3 KO mice infected with SARS-CoV-1 showed significantly reduced systemic inflammation compared to the wild-type controls [19]. At that time, it was suggested that inhibition of the complement system could be an effective treatment option of the inflammation caused by the coronavirus infection. At the outbreak of the COVID-19 pandemic, the importance of the complement system became more apparent [20], and several research groups conducted intensive research concerning the mechanism of SARS-CoV-2-induced complement activation. Clinical evidence suggested that the LP might be an important contributor to the acute respiratory failure and systemic coagulopathy that have pivotal roles in the morbidity and mortality of severe COVID-19. Deposited MASP-2, a key protease of the lectin pathway, was detected in the lungs and kidneys of patients who died from COVID-19 [6,7,8]. However, the mechanism through which the LP is activated in the course of SARS-CoV-2 infection is far from clear. There is a consensus in the literature that the S protein, which is heavily glycosylated, binds MBL and activates the LP [10,11]. The findings about the contribution of the most abundant viral structural protein, N protein, are controversial. There are publications claiming that N protein binds PRMs and elicits LP activation [10,13]. Other researchers could not support these findings [11,12]. Also, there are publications claiming that N protein binds directly to the MASP-2 protein and potentiates its enzymatic activity, facilitating pathological LP overactivation [13,14]. The source of the contradictory results could be inadequate experimental settings, and in the first place, the different quality (purity and activity) of the recombinant proteins used in the studies. Different authors used recombinant proteins from various sources (in-house made, commercially purchased, produced in different hosts) [13,14]. As we have decades of experience in expressing biologically active recombinant proteins, particularly lectin pathway proteases [18,21,22,23,24], we decided to examine the role of the N protein in lectin pathway activation.

First, we examined whether deposited N protein can activate the LP in normal human serum. It could be of great importance since the N protein molecules, released from the infected cells, can nonspecifically stick to any surface, e.g., to the surface of the endothelial cells, and initiate complement activation. The N protein is 419 amino acids long, its molecular weight is 46.6 kDa, and it has two structural domains: the RNA-binding N-terminal domain (NTD) and the C-terminal domain (CTD) [25]. Between the NTD and CTD, there is a serine/arginine-rich intrinsically disordered linker region. The NTD binds to the 3′-end of the viral RNA, but the unstructured regions containing positively charged amino acids also contribute to the interaction with the negatively charged nucleic acid through electrostatic interactions. It means that the N protein binds nucleic acid with high affinity, and special attention must be paid during purification of the recombinant protein to remove the nucleic acid contamination. On the other hand, we cannot exclude that the N protein in the circulation carries nucleic acids of viral or human origin. Therefore, we tested two types of recombinant N protein preparations isolated from *E. coli* cells, nucleic acid-free and nucleic acid-contaminated proteins. We found that the nucleic acid-free N protein did not activate the LP or the CP since there was no C4b deposition. The nucleic acid-containing N protein elicited a very weak C4b deposition that could not be abolished by LP inhibitors (SGMI-1 or SGMI-2); therefore, it can be considered as a weak CP activation. As for the C3 activation, N protein triggered strong C3b deposition in 10% NHS. We can conclude that pure surface-deposited N protein cannot initiate LP or CP activation, but it can activate the AP. Unlike the other structural proteins of SARS-CoV-2, the N protein is located in the nucleocapsid and does not go through the secretory pathway; therefore, it is not glycosylated [26]. However, if recombinant N protein is expressed in mammalian cells (e.g., HEK293), and it is forced through the secretory pathway by fusing a leader sequence to its N-terminus, it becomes glycosylated. We have checked the LP activating ability of such artificially glycosylated N protein, but we could not detect significant C4b deposition.

We checked the ability of N protein to boost the enzymatic/proteolytic activity of MASP-2. Our recombinant MASP-2 protein (catalytic fragment containing the CCP1-CCP2-SP domains) showed high catalytic activity on a synthetic substrate, which could not be further enhanced by addition of N protein even at very high concentration. The MASP-2 is present as a zymogen in the circulation before LP activation occurs. The zymogen form of the trypsin-like proteases may have some enzymatic activity, which might be important during the proteolytic autoactivation process [15]. To check whether the N protein can stimulate the enzymatic activity of the one-chain zymogen MASP-2, we incubated the Arg444Gln zymogen mutant MASP-2 with N protein. We did not detect any cleavage of the synthetic substrate, which means that the N protein cannot stimulate the enzymatic activity of the zymogen form of MASP-2. Our results are in contrast to those of Kang et al., whose recombinant MASP-2 is obviously inactive (its enzymatic activity is practically equal to that of the control protein), and very likely their recombinant N protein is contaminated with some protease [14]. The N protein concentrations at which they obtained significant effect are extremely high (8–16 µM), considering that the average concentration of the viral nucleoprotein in the blood in case of infection is only about 2 × 10^−5^ µM (1000 pg/mL) [27]. Even if we consider the highest blood concentration of N protein measured in COVID-19 patients (1 × 10^5^ pg/mL = 2 × 10^−3^ µM), the concentration they used exceeds that value several thousand times. It is highly unlikely that the N protein at the average plasma concentration (1000 pg/mL) would influence the enzymatic properties of MASP-2.

We also tested the effect of the N protein on the proteolysis of C4. MASP-2 is extremely efficient at cleavage of the C4 substrate [28]. We have shown earlier that even the one-chain zymogen form of MASP-2 (Arg444Gln mutant) is capable of cleaving C4 [15]. In our experiment, the recombinant MASP-2 protein cleaved C4 very efficiently, which could not be enhanced further by the addition of N protein. It is true for both the wild-type and the stable zymogen MASP-2 mutant, which means that the N protein does not facilitate the conformational change between the zymogen and the active form. On the other hand, during these experiments we noticed that the MASP-2 protease digests the N protein. It is not surprising, taking into account that the N protein contains unstructured regions with easily accessible arginine residues for the trypsin-like protease. It is very likely that other proteases in the blood (other complement and coagulation system proteases, etc.) can also cleave the N protein. The fact that the N protein in the circulation can be easily degraded does not support the theory that it would make a significant impact on the LP activation.

We also tested the effect of N protein in an assay where the MBL–MASP complexes were deposited from the serum to a mannan-coated surface. This scenario is closer to the physiological situation than the experiments that use isolated proteins. In the presence of high salt concentration (1 M NaCl), C1q cannot bind to immune complexes and the C1 complex dissociates, while the MBL–MASP complexes remain intact and through MBL they bind to the activator surface [17]. In such conditions, therefore, there is no C1 on the surface and the only enzyme capable of cleaving C4 will be MASP-2. After washing, C4 was added to the plates with or without N protein. The deposited MBL–MASP-2 complexes cleaved the C4 component; however, the presence of the N protein did not make any difference in the extent of C4 cleavage. We can draw the conclusion that the N protein cannot influence the enzymatic properties of MASP-2 even in the case when MASP-2 is in complex with MBL.

In the previous studies where the effect of the N protein on the properties of MASP-2 was investigated, the role of MASP-1 was not taken into account [10,13,14]. MASP-1 is the exclusive physiological activator of MASP-2 in the NHS [3,29,30]. It is possible that the N protein induces MASP-2 (auto)activation in the MBL–MASP complexes, but we could not detect it because the activity of MASP-1 overrides it. To check this possibility, we added our MASP-1-selective inhibitor, SGMI-1, into NHS and detected the C4b deposition on mannan-coated plates. The inhibition of MASP-1 completely prevented LP activation and the addition of N protein could not undo this inhibition. We have previously proved that in vivo zymogen MASP-2 cannot auto-activate. This experiment proved that the presence of N protein does not make any difference; without the contribution of MASP-1, zymogen MASP-2 still cannot initiate LP activation.

We also tested the direct interaction between MASP-2 and the N protein using BLI. We immobilized N protein on the sensor while the interacting partners were in the fluid phase. As it was expected, the N protein bound PTX3 with high affinity [11]. This proved that our recombinant N protein preparation contained N protein in native conformation. We could not, however, detect any binding interaction between MASP-2 and the N protein. We tested the zymogen, the active, and the inactive forms of MASP-2. The only interaction we could detect was the MASP-2-mediated proteolysis of the N protein. Our results are in sharp contrast with that of Gao et al., who used a similar method, SPR [13]. They did not disclose, however, which was the immobilized protein on the sensor chip and what method (i.e., immobilization technique) they used. The lack of this information makes it difficult to compare the results obtained with the two methods (SPR and BLI). We also tested whether the N protein binds MASP-1. Since we could not detect any interaction between the two proteins in our BLI experiments, we can rule out the possibility that the N protein could influence the activity of MASP-2 through its physiological activator, MASP-1. The results obtained from BLI measurements were further verified through ELISA assays, which tracked the binding of MASP-2 to immobilized N protein. However, no specific binding of MASP-2 to the N protein was observed, even at concentrations of 1, 2, 3, and 10 µg/mL (Appendix A).

## 4. Materials and Methods

### 4.1. Recombinant Proteins

The synthetic gene encoding the full length nucleocapsid protein of SARS-CoV-2 was purchased from Thermo Fisher Scientific (Waltham, MA, USA) according to GenBank sequence QHD43423.2. It was subcloned into periplasmic pET-20b(+) plasmid vector using the forward primer 5′ GCG CC ATGG CG CATCACCATCACCATCACG 3′ that contained the 6xHis tag at N terminus. The applied reverse primer was 5′ GTG GAG GCGGCCGC TTA 3′. Competent *E. coli* BL21(DE3) pLysS cells were transformed with plasmid and were grown at 37 °C. The protein expression was induced with 100 µg/mL IPTG at optical density 0.8. Temperature was reduced to 16 °C and cells were harvested after overnight incubation and were resuspended in PBS.

The purification of nucleic acid-free N protein was performed as described by Carlson et al. with some minor modifications [31]. N protein was purified under denaturing conditions in the presence of 6 M urea to eliminate all bound nucleic acids. Frozen cells from 1 L cell culture were thawed and suspended in 1/6 volume of carrier buffer for Ni-affinity chromatography (50 mM HEPES pH 7.5, 500 mM NaCl, 10% glycerol, 6 M urea, 20 mM imidazole). Cells were sonicated for 12 min on ice. The suspension was centrifuged for 1 h at 43,000 g, and the supernatant was collected, filtered, and applied onto a Ni-NTA Superflow chromatography column. The elution was performed using a linear gradient of 20–250 mM imidazole. To eliminate urea, N protein-containing fractions were collected, concentrated on 30 kDa filter, and dialyzed against 50 mM HEPES pH 7.5, 500 mM NaCl, 10% glycerol buffer (storage buffer). The final purification step was size-exclusion chromatography on Superdex 200 (Cytiva, Marlborough, MA, USA) preparative column in storage buffer. Nucleic acid content was determined as the ratio of absorbances at 260 nm and 280 nm. The value of 260/280 was 0.5 or below in the case of denaturing purification indicating that the N protein preparation resulted in a nucleic acid-free sample.

Nucleic acid-contaminated N protein was purified in similar method except that the denaturation step was omitted. Resuspension of cells was performed in urea-free carrier buffer of Ni-affinity chromatography, which contained 50 mM HEPES pH 7.5, 500 mM NaCl, 5 mM β-mercaptoethanol, 20 mM imidazole, and 10% glycerol. Eluted fractions were concentrated on 30 kDa filter and introduced to the final SEC on Superdex 200 column in storage buffer. The 260/280 ratio of nucleic acid-contaminated N protein preparation was over 1.2.

In our experiments, the CCP1-CCP2-SP domains, also known as the catalytic fragments of the MASP enzymes, were used. They possess the same enzymatic properties as their full-length counterparts. The wild-type catalytic fragment of MASP-2, the catalytically inactive Ser633Ala mutant (precursor numbering), the Arg444Gln mutant, and also the MASP-1 Ser646Ala mutant were recombinantly expressed in *E.coli*; inclusion bodies were renatured and purified as published earlier [16,18,21,22,23]. The enzymatic activity of all prepared MASP-2 CCP1-CCP2-SP samples was verified on synthetic Z-Lys-S-Bzl substrate and their complex-forming ability with C1 inhibitor was also tested. MASP-2 CCP1-CCP2-SP Arg444Gln mutant is a one-chain, non-autoactivating zymogenic form of MASP-2. The MASP-1 Ser646Ala mutant and the MASP-2 Ser633Ala mutant were proteolytically cleaved in their activation peptide segment. The resulting catalytically inactive two-chain forms, nevertheless, accommodate an active-like conformation.

### 4.2. Other Samples

Human complement component C4 was purchased from Complement Technology Inc. (Tyler, TX, USA) (#A105c). Human pentraxin-3 is a product of Sino Biologicals (Beijing, China) (#12082-H08H). MASP-1 and MASP-2 inhibitors SGMI-1 and SGMI-2, respectively, were recombinantly expressed in *E.coli* Shuffle T7 cells and purified as described earlier [32].

Normal human serum (NHS) was pooled from ten healthy volunteers. Blood samples were collected by venipuncture using Sarstedt S-Monovette Z tubes (Nümbrecht, Germany) (#02.1063). After 30 min clotting at room temperature, tubes were centrifuged at 1500 g for 15 min at room temperature. Sera were combined and were snap-frozen in small aliquots.

### 4.3. ELISA

Several types of ELISA assays were applied in our experiments. The activation of complement pathways was triggered on different surfaces and complement product depositions from NHS were detected.

C4b and C3b deposition assays were detailed in a previous paper [33]. Briefly, LP activation was initiated on 10 µg/mL mannan-coated high-binding microtiter plates (Greiner Bio-One, (Mosonmagyarovar, Hungary) #655061), and 2% normal human serum was applied in 10 mM HEPES pH 7.5, 150 mM NaCl, 5 mM CaCl_2_, 5 mM MgCl_2_, 0.05% Tween-20 serum dilution buffer in the presence or absence of 2 µM N protein. The deposited C4b was detected with 1/2000-time diluted anti-human C4c antibody (DAKO, (Santa Clara, CA, USA) #Q0369).

The complement activation on N protein-coated surface was performed as follows. Nunc Immuno Maxisorp microtiter plate (Thermo Fisher Scientific #442404) was coated with 10 µg/mL nucleic acid-free or nucleic acid-containing N protein diluted in 15 mM Na_2_CO_3_, 35 mM NaHCO_3_, pH 9.6 overnight at 4 °C. Wells were blocked with 1% BSA (bovine serum albumin) dissolved in TBS buffer (Merck Sigma-Aldrich, Darmstadt, Germany, #94158). After washing the plate three times with TBS supplemented with 0.1% Tween-20 and 5 mM CaCl_2_, 10% NHS diluted in serum dilution buffer was added and incubated for 1 h at 37 °C. In order to eliminate the activity of MASP-1 or MASP-2, 20 µM of their specific inhibitors, SGMI-1 or SGMI-2, respectively, was added to NHS prior to incubation. The plate was washed again, and then C4b or C3b depositions were detected by 1/2000-time diluted anti-human C4c antibody or 1/5000-time diluted anti-human C3c antibody (DAKO, #A0062), respectively. Antibodies were diluted in TBS, 1% BSA, 5 mM CaCl_2_, 0.1% Tween-20 buffer and were incubated for 1 h at 37 °C. Followed by the next washing step, anti-rabbit-HRP secondary antibody (Sigma-Aldrich, #A1949) was introduced in 1/40,000 dilution for 30 min at 37 °C. Detection was carried out with 100 µg/mL TMB substrate dissolved in 50 mM citrate–phosphate buffer pH 5.5 in the presence of 0.1% H_2_O_2_.

The effect of N protein on proteolytic activity of MBL–MASP-2 in NHS was examined in an experiment introduced by Petersen et al. [17], but with some modifications. In essence, MBL–MASP-2 complexes from 2% NHS were deposited in mannan-coated wells. Classical pathway components, such as C1q, C1r, and C1 were washed away from the surface with high-ionic-strength buffer, 100 ng/well purified C4 in the absence or in the presence of 250 nM N protein was added to the wells, and deposition of the cleavage product C4b was followed over time as described above.

Interaction between N protein and MASP-2 was assessed by direct binding assays in ELISA measurements. Microtiter plates were coated with 10 µg/mL N protein, 10 µg/mL wild-type MASP-2 (CCP1-CCP2-SP) as positive control and 10 µg/mL BSA as negative control surface overnight at 4 °C. Wells were blocked with 1% BSA in TBS buffer for 2 h at 37 °C. Wild-type MASP-2 (CCP1-CCP2-SP) was added at 1, 2, 3, and 10 µg/mL concentration in serum dilution buffer supplemented with 1% BSA and wells were incubated for 1 h at 37 °C. The presence of bound MASP-2 was followed by anti-MASP-2 polyclonal antibody (Sigma-Aldrich, #HPA029313). Visualization by TMB substrate was performed as mentioned earlier.

### 4.4. Enzymatic Activity Assays

The possible effect of N protein on the enzymatic activity of MASP-2 was investigated on a synthetic peptide substrate and on its natural substrate, C4 using purified components.

Experimental conditions were set up for activity measurements on the synthetic peptide substrate as described by Kang et al. [14]. The only difference was the substrate itself; we used another sensitive substrate for MASP-2, namely Z-Gly-Gly-Arg-AMC (Bachem, Budendorf, Switzerland, 4002155). Briefly, 320 nM MASP-2 was preincubated with 16 µM N protein or its storage buffer. The reaction buffer contained 50 mM HEPES pH 7.5, 150 mM NaCl, 0.2% PEG-8000, and 3.5% glycerol. After 10 min, serial dilution of Z-Gly-Gly-Arg-AMC substrate was added. The cleavage of substrate was followed on PerkinElmer EnSpire spectrofluorometer (Shelton, Connecticut, USA) for 10 min at 37 °C at 380 nm/460 nm excitation and emission wavelengths, respectively. Initial velocities were calculated by linear regression, and they were plotted against substrate concentration. To measure the activity of the zymogen MASP-2, 4.5 µM MASP-2 Arg444Gln mutant was preincubated with 16 µM N protein. The cleavage of 50 µM Z-GlyGly-Arg-AMC was followed for 15 min. In the control experiment, 16 µM N protein was incubated and measured with the substrate alone under the same conditions.

The proteolytic activity of active and zymogen MASP-2 can be measured on its natural substrate C4. Two nM MASP-2 was incubated with 1 µM purified C4 and 4 µM N protein or the same volume of its storage buffer. The reaction buffer contained 10 mM HEPES pH 7.5, 150 mM NaCl, 5 mM CaCl_2_, and 3% glycerol. The enzymatic reactions were incubated at 37 °C; samples were taken at 0 min, 20 min, and 60 min and were run on SDS-PAGE under reducing conditions. C4 was incubated with the same amount of N protein for 60 min to visualize possible self-degradation. In the case of zymogen MASP-2, 3 µM of Arg444Gln mutant was applied using the same experimental setup; samples were taken at 0 min, 45 min, and 90 min. Appearance of the cleaved α’ chain and the disappearance of the α chain were analyzed with densitometry and expressed as percentage of the non-cleaved C4.

To assess enzymatic activity of MASP-2 towards N protein, 750 nM wild-type catalytic fragment of MASP-2 was incubated with 2.8 µM N protein for 60 min at 37 °C. Samples were taken at different timepoints, and the cleavage was stopped by adding reducing SDS-PAGE sample buffer containing 5% β-mercaptoethanol. Two control reactions, N protein alone and N protein with MASP-2 and 10 µM of its inhibitor, SGMI-2, were also incubated in the same conditions for 60 min to assess specific cleavage. Samples were run on SDS-PAGE and visualized by Coomassie Brillant Blue staining.

### 4.5. Bio-Layer Interferometry (BLI)

Sartorius Octet^®^ BLI K2 instrument was used to detect the interaction between MASP-2 and N protein. Experimental conditions were set up according to the manufacturer’s instructions. Prior to the experiment, anti-penta-His HIS1K biosensors (Sartorius, Goettingen, Germany #18-5120) were pre-wetted in HBS buffer (20 mM HEPES pH 7.5, 150 mM NaCl). The temperature was kept at 30 °C, shaking speed was set at 1000 rpm, and 200 µL volumes were used in all wells. Optimal loading conditions were experimentally determined applying nucleic acid-free N protein as ligand. HBS-T buffer (20 mM HEPES pH 7.5, 150 mM NaCl, 0.05% Tween-20) was applied in wells dedicated to baseline and dissociation steps. The analytes were diluted also in HBS-T buffer. To rule out non-specific interactions, two types of control sensors were applied: one without ligand and one without the analyte. Association and data of the examined analyte were calculated subtracting the corresponding data collected from the “no ligand” sensor.

Due to its extremely high positive charge, N protein was prone to attach very strongly to the sensor; therefore, a regeneration step was carried out in 100 mM glycine, pH 2 for 5 s, repeated four times. The experimental sequence setup used started with a regeneration/neutralization step, followed by baseline step in HBS buffer for 60 s. Ligand loading was set to 50 s in 5 µg/mL N protein solution. A second baseline step of 180 s in HBS-T preceded the kinetic analysis involving the association step for 360 s and the dissociation step for 180 s.

Our experimental setup was validated by human pentraxin-3, an interaction partner of N protein known from the literature [11]. Serial dilution of pentraxin-3 was applied as analyte, and K_D_ value was determined assuming 1:1 binding model using ForteBio DataAnalysis 10.0.1.7 software. Since active MASP-2 possesses proteolytic activity on N protein, both the wild-type enzyme and also the zymogen Arg444Gln mutant were unsuitable for the measurements. Therefore, in the case of MASP-2, catalytically inactive but cleaved two-chain Ser633Ala mutant was used in BLI experiments. In the control experiment, the similar form of MASP-1 Ser646Ala mutant was used.

## 5. Conclusions

Sharing negative results should be a general interest in the scientific community as it would save a tremendous amount of time and energy for fellow researchers. It is even more important to do so when the negative results disprove previously published statements related to a topic of great medical relevance.

In conclusion, we found that the N protein does not make any detectable interaction with MASP-2, and it does not stimulate its enzymatic activity. It is true for the recombinant catalytic fragment and also for the full-length serum protein. The N protein could not facilitate the (auto)activation of MASP-2, which was completely MASP-1-dependent in NHS. The N protein could not enhance LP activity in any way. MASP-2 degrades the N protein; however, the biological relevance of this phenomenon is not clear. The N protein triggers AP activation, which might have physiological significance. Since the N protein is an efficient immunogen, it likely activates the CP through anti-N protein antibodies in the sera of infected individuals, but it certainly does not activate the LP.

## Figures and Tables

**Figure 1 ijms-25-07343-f001:**
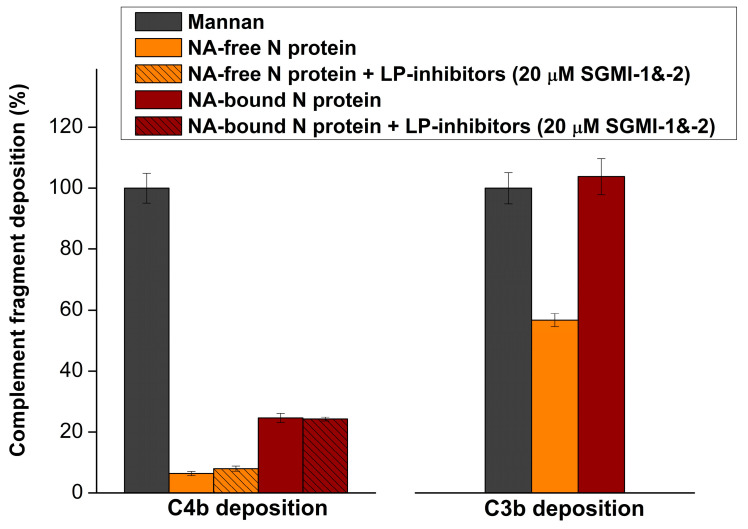
Complement activation on nucleic acid-containing (NA-bound) and nucleic acid-free (NA-free) nucleocapsid (N) protein samples. Complement activation products C3b and C4b deposited from 10% normal human serum on the surface of nucleic acid-free (orange columns) and nucleic acid-containing (brown columns) N protein were detected. Mannan-coated surface served as positive control for lectin pathway activation and is marked as 100% deposition of complement fragments. The presence of nucleic acid bound to N protein resulted in elevated complement activation that could not be attenuated by lectin pathway (LP) inhibitors (striped columns). The presence of Ca^2+^ and 10% serum concentration allowed all three pathways to be initiated on the surface of N protein, resulted in strong C3b deposition. C4b deposition can occur only through lectin and/or classical pathway activation.

**Figure 2 ijms-25-07343-f002:**
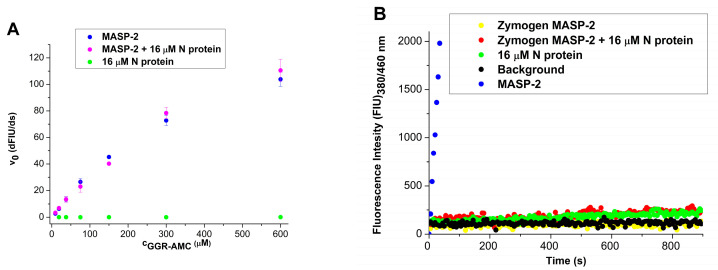
The effect of N protein on the enzymatic activity of mannose-binding lectin-associated serine protease-2 (MASP-2) on synthetic peptide substrate. (**A**) Initial velocity of Z-Gly-Gly-Arg-AMC cleavage was plotted against the substrate concentrations. MASP-2 catalytic fragment was used in 320 nM concentration in the presence (pink) and in the absence (blue) of 16 µM N protein. N protein alone was also added to the substrate (green). (**B**) The activity of zymogenic MASP-2 Arg444Gln mutant (4.5 µM) on 50 µM Z-Gly-Gly-Arg-AMC was followed in the presence (red) and in the absence (yellow) of 16 µM N protein. The substrate was also incubated with 16 µM N protein alone (green). The activity of 320 nM wild-type MASP-2 on 50 µM Z-Gly-Gly-Arg-AMC is indicated in blue.

**Figure 3 ijms-25-07343-f003:**
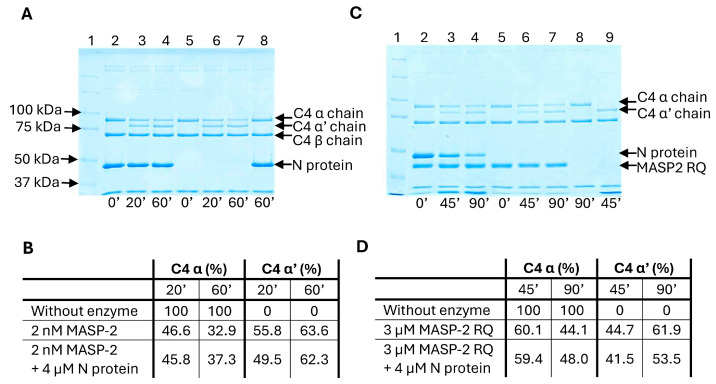
The effect of N protein on the proteolytic activity of wild-type MASP-2 (**A**,**B**) and zymogen MASP-2 Arg444Gln (RQ) mutant (**C**,**D**) on complement C4 protein. C4 was incubated with 2 nM wild-type MASP-2 or with 3 µM zymogen MASP-2 Arg444Gln mutant in the presence or absence of 4 µM N protein at 37 °C for different times indicated under the gels. Samples were run under reducing conditions and gels were stained with Coomassie Brillant Blue dye. (**A**) Lane 1, Kaleidoscope molecular weight standard; lanes 2–4, C4 cleavage in the presence of 4 µM N protein; lanes 5–7, C4 cleavage without N protein; lane 8, intact C4 incubated for 90 min with N protein. (**B**) Band quantification of C4 cleavage by MASP-2: bands were analyzed with densitometry. (**C**) Lane 1, Kaleidoscope molecular weight standard; lanes 2–4, C4 cleavage in the presence of 4 µM N protein; lanes 5–7, C4 cleavage without N protein; lane 8, intact C4 incubated for 90 min; lane 9, cleaved C4. (**D**) Band quantification of C4 cleavage by zymogen MASP-2 Arg444Gln mutant: bands were analyzed with densitometry.

**Figure 4 ijms-25-07343-f004:**
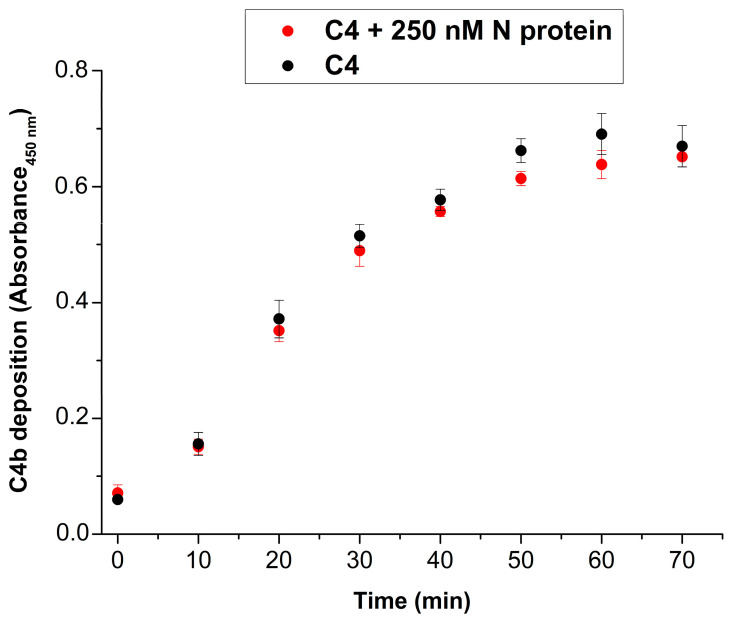
The effect of N protein on C4 cleavage ability of MBL–MASP-2 complexes. MBL–MASP-2 complexes were deposited on microtiter plate from normal human serum. After washing the wells with high-ionic-strength buffer that removed components of the classical pathway (C1q, C1r, C1s), C4 was added in the presence (red) or in the absence (black) of N protein. The deposition of C4b was followed over time.

**Figure 5 ijms-25-07343-f005:**
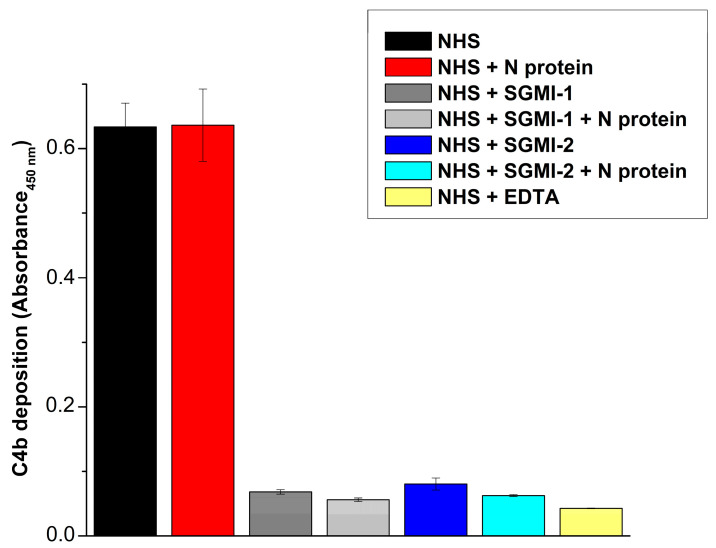
The effect of N protein on lectin pathway activation in normal human serum under MASP-1 or MASP-2 inhibition. Lectin pathway activation was initiated on mannan-coated microtiter plate. C4b deposition was followed in 2% NHS in the presence or absence of 2 µM N protein. LP-specific inhibitors (SGMI-1 or -2) were also added at 20 µM concentration. Activation of MASP-2 by MASP-1 was completely blocked with our MASP-1-specific inhibitor, SGMI-1. In these conditions, any rise in the rate of zymogen MASP-2 autoactivation resulting from the presence of N protein could be detected. However, N protein had no such effect.

**Figure 6 ijms-25-07343-f006:**
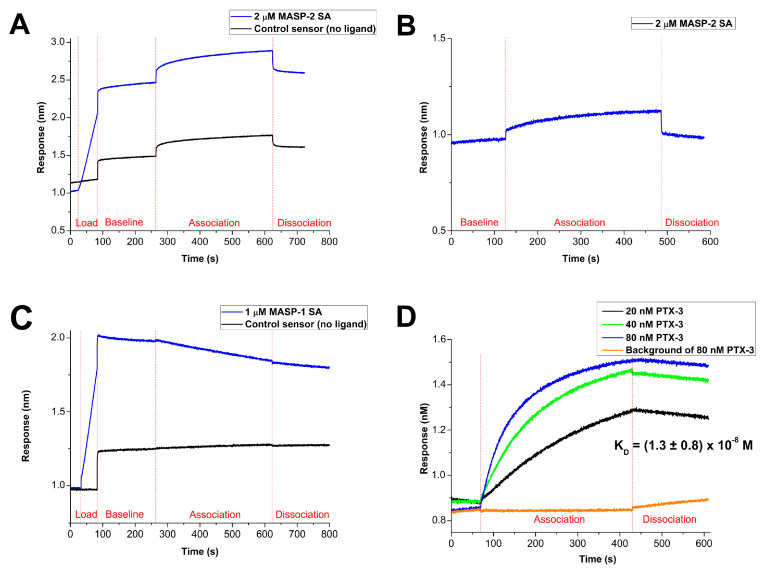
The interaction between N protein and MASP-2 (**A**,**B**), MASP-1 (**C**), and pentraxin-3 (**D**). Bio-layer interferometry (BLI) measurements were carried out on Sartorius Octet^®^ BLI using His1K sensor. N protein was immobilized as ligand, while MASP-2, MASP-1, and pentraxin-3 served as analytes. MASP-1 and -2 were catalytically inactive Ser-Ala mutants. These mutants were cleaved in the activation peptide (Arg-Ile) by limited proteolysis. The resulting two-chain forms possessed an active-like conformation although they remained catalitically inactive [16,18]. (**A**) Two µM MASP-2 Ser633Ala (SA) (blue line) did not show interaction with N protein compared to the control sensor (black line), where no ligand was immobilized. (**B**) Subtracting the control measurement, only a weak, non-specific binding was observed. (**C**) One µM MASP-1 Ser646Ala (SA) mutant (blue line) did not present specific association compared to control (black line). (**D**) Pentraxin-3 validated our experimental system demonstrating strong concentration dependent interaction with N protein. K_D_ value is an average of measurments at five different pentraxin-3 concentrations (5, 10, 20, 40, 80 nM). Standard deviation is indicated in the figure.

## Data Availability

The datasets generated during and/or analyzed during the current study are available from the corresponding author on reasonable request.

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
