# Peer review of "SARS-CoV-2 Nucleocapsid Protein Is Not Responsible for Over-Activation of Complement Lectin Pathway"

_ijms, 2024, doi:10.3390/ijms25137343_

Round 1

Reviewer 1 Report

Comments and Suggestions for Authors

The manuscript “SARS-CoV-2 nucleocapsid protein is not responsible for the over-activation of complement lectin pathway” (manuscript ID: ijms-3043393) refers to a thorough investigation of the role that the nucleocapsid (N) protein of SARS-CoV-2 has been proposed to play in complement activation through the lectin pathway. The theoretical background of the study has concisely been presented in the introduction part. The materials and methods used during the experimental investigation have been described in a clear and detailed way. The results obtained, although being contradictory with some previously reported data, have been presented and discussed in a clear, straightforward, and convincing way. Overall, at least in my opinion, the manuscript deserves publication, since it might be part of a fruitful debate among groups working in the field and will contribute to further elucidating whether and how N protein of SARS-CoV-2 may affect complement activation during COVID-19 infection.

Minor comments

-Line 101: (Ali) should be omitted.

-Figure 3C and Figure 3D are not mentioned in the text. I can guess that: (Figure 3A), line 156, should be replaced with (Figure 3A, Figure 3B), while (Figure 3B), line 157, should be replaced with (Figure 3C, Figure 3D). Is it so?

Line 238: “1.8 * 10-8 M” might be better to appear as “1.8 x 10-8 M”.

-Lines 257-259: “MASP-1 and -2 were catalytically inactive SA mutants, but they were cleaved in the activation peptide and possessed their two-chain forms the presumes an active-like conformation”: the meaning is not quite clear.

-Line 297: It would be useful to provide the molecular weight of the N protein.

-Line 383: The number of the reference corresponding to Gao et al., should be provided, i.e., [13].

-Figure 4, Figure 5, Figure 6, Figure S1: In the Y-axis, the “,” symbol should be replaced with the “.” symbol, e.g., “0.2” should appear instead of “0,2”.

-Full terms for “MBL”, “NHS”, “SA mutants” etc., should be provided upon first appearance of the corresponding abbreviated form.

-It might be better to use either the single-letter or the three-letter code for amino acids throughout the text.

-For uniformity reasons, it might be better to use either L/mL or l/ml throughout the text.

Reviewer 2 Report

Comments and Suggestions for Authors

Authors clarify the role of the N-protein in the lectin pathway activation they decided to examine its capacity to activate the lectin pathway and its interaction with lectin pathway proteases MASP-1 and MASP-2 to reveal the activation mechanism.

Although this manuscript is potentially interesting, several issues arise.

1)    This study showed negative results. Therefore, this manuscript may not be interesting for the readers.

2)    What is responsible for the strong lectin pathway activation elicited by SARS-CoV-2 infection?

3)    Conclusion in the last of discussion may be helpful for the leaders.

4)    What is abbreviation, LP in Figure legend?

5)    The abbreviation of LP is not clear. Authors should use lectin pathway.

6)    Abstract should be further explained?

7)    There are many abbreviations in Text and Figure legends. These abbreviations should be explained.

8)    It may be further interesting that authors emphasized positive data.

Reviewer 3 Report

Comments and Suggestions for Authors

In this manuscript, the authors want to confirm whether the N protein of SARS-CoV-2 could promote the lectin pathway of the host complement system. The main reason was that the results of other studies indicated that the N protein of SARS-CoV-2 would promote the activation of the lectin pathway by binding to MASP-2, but there were many controversial data in these studies. Therefore, the authors further investigated the relationship between the N protein and the alternative pathway, and found that the N protein neither interact with MASP-1 and MASP-2, nor promote the activation of MASP-2. Instead, the N protein activate the alternative pathway, which is important for the subsequent development of SARS-CoV-2 and the alternative pathway. On the contrary, it promotes the alternative pathway, which is helpful for the subsequent study on the complement system activation by SARS-CoV-2 N proteins. However, the results there are some issues that need to be clarified, such as: 1.

1. The effect of N protein on the MASP-2 enzyme activity is negative. As we might wondering if there is any “positive” control for this experiment?

2. Regarding the experiment on the interaction of N protein with MASP-1 and MASP-2 (BLI experiment), (1) the result of the graph only shows the Kd value of PTX3 and N protein, but not the Kd value of MASP-1 and MASP-2 and N protein, is it possible to show this data? (2) Regarding the Kd values of PTX3 and N protein, please explain which concentration of PTX3 interacts with N protein. (3) I would like to ask if the negative signals of N protein and MASP-2 in BLI experiments described in this paper are due to the fact that MASP-2 hydrolyzes N protein, and what is the basis for this conclusion, please explain in detail.

Round 2

Reviewer 2 Report

Comments and Suggestions for Authors

This manuscript has been sufficiently improved. I have no furhter comment. 

Reviewer 3 Report

Comments and Suggestions for Authors

1.     In question 2, the authors answered that it is not possible to determine whether the N protein interacts with MASP-1,-2, but in lines 232 and 233 of this revised manuscript, there are reports that there is a direct interaction between the MASP-2 catalytic fragment (CCP1-CCP2-SP) and the N protein, so I would like to ask why there is such a big discrepancy. 

2. Please employ other experimental methods, such as CO-IP or pull-down assays, to confirm that the N protein doesn't interact with MASP-1,-2.

3.  In question 2, the author's answer is to average the Kd values of PTX3-N protein interactions at 5 different concentrations (5, 10, 20, 40, 80 nM), and I would like to ask why it is necessary to average the Kd values instead of displaying the Kd values of PTX3-N protein interactions at each concentration, please answer this question.

Round 3

Reviewer 3 Report

Comments and Suggestions for Authors

I do not have any further comment with regard to the revised manuscript.